

# Comparison of calibration characteristics of different acoustic impact systems for measuring bedload transport in mountain streams

Dieter Rickenmann[1], Lorenz Ammann[1], Tobias Nicollier[1], Stefan Boss[1], Bruno Fritschi[1], Gilles Antoniazza[2,1], Nicolas Steeb[1], Zheng Chen[1,3,4], Carlos Wyss[1,5], Alexandre Badoux[1]

[1]Swiss Federal Research Institute WSL, Birmensdorf, 8903, Switzerland
[2]Institute of Earth Surface Dynamics (IDYST), University of Lausanne, Lausanne, 1015, Switzerland
[3]Institute of Mountain Hazards and Environment, Chinese Academy of Sciences, Chengdu, 610041, China
[4]University of Chinese Academy of Sciences, Beijing, 100049, China
[5]wyss.io - Dr. Carlos R. Wyss Engineering, Zürich, 8052, Switzerland

*Correspondence to*: Dieter Rickenmann (dieter.rickenmann@wsl.ch)

**Abstract.** The Swiss plate geophone (SPG) system has been installed and tested in more than 20 steep gravel-bed streams and rivers, and related studies generally resulted in rather robust calibration relations between signal impulse counts and transported bedload mass. Here, we compare this system with three alternative surrogate measuring systems. A variant of the SPG system uses the same frame (housing) set-up but with an accelerometer instead of a geophone sensor to measure the vibrations of the plate (GP-Acc, for geophone plate accelerometer). The miniplate accelerometer (MPA) system has a smaller dimension of the impact plate and is embedded in more elastomer material than the SPG system. The Japanese pipe microphone (JPM) is a 1 m long version of the system that has been installed in many streams in Japan. To compare the performance of the four systems, we used calibration measurements with direct bedload samples from three field sites and an outdoor flume facility with controlled sediment feed. At our field sites, the systems with an accelerometer and a microphone showed partly large temporal variations in the background noise level, which may have impaired the calibration measurements obtained during certain time periods. Excluding these periods, the SPG, GP-Acc and JPM all resulted in robust calibration relations, whereas the calibration of the MPA system showed a poorer performance at all sites.

## 1 Introduction

The measurement of bedload transport in steep streams is a difficult task (e.g. Gray et al., 2010; Rickenmann, 2017a). This is particularly true regarding direct measurements that typically have a limited resolution in space and time, and that are sometimes challenging to achieve for higher streamflow and transport conditions. Direct bedload sampling includes the use of retention basins, slot samplers or mobile basket samplers (e.g. Helley and Smith, 1971; Gray et al., 2010), limited by factors such as sampler capacity (e.g. Habersack et al., 2017; Nicollier et al., 2021), flow conditions (e.g. Bunte et al., 2004) or bed material texture (Camenen et al., 2012). Physical traps and samplers provide a sample of bedload particles transported





into the measuring devices during a known period. While some methods allow to collect the entire grain size distribution of the bedload particles transported over the entire stream width, other methods only sample fractions some of the transported bedload or grain sizes (Aberle et al., 2017).

To overcome some of the limitations associated with direct bedload measurements in steep streams, increasing efforts were made in the last decades to apply and test "indirect" or surrogate monitoring techniques (e.g., Rickenmann, 2017a, 2017b; Gimbert et al., 2019; Geay et al., 2020) that particularly include passive acoustic measurements of bedload transport. These systems essentially record naturally generated noise signals, i.e. the sound or vibration induced by moving bedload particles. A critical advantage of indirect monitoring techniques lies in their ability to record a bedload signal continuously in time,

including during flood flows, and this over an entire channel cross-section to provide detailed spatial information. However, the calibration of passive acoustic measurements typically requires concurrent sampling measurements of bedload transport, preferably collected at the same field site where the passive acoustic sensors are deployed.

Passive acoustic or seismic monitoring techniques include the hydrophone, i.e. an underwater microphone (Thorne, 1986, 2015; Geay et al., 2017, 2020), the Japanese pipe microphone (JPM) Mizuyama et al., 2010a, b; Mao et al., 2016; Tsutsumi

et al., 2018; Choi et al., 2020), the impact plate (e.g. Rickenmann and McArdell, 2007; Krein et al., 2008; Raven et al., 2010; Hilldale et al., 2015; Wyss et al., 2016a; Kuhnle et al., 2017; Koshiba et al., 2018), and the seismometer (Roth et al., 2016; Dietze et al., 2019; Gimbert et al., 2019, Bakker et al., 2020). The main sources of sediment-related noise detected by hydrophones are inter-particle collisions among moving particles or between moving particles and the streambed (Thorne, 1986, 2015). The measuring devices that record the impacts of sediment particles onto a (metallic) structure (i.e. a pipe, a

plate, or a cylinder), use further types of sensors (i.e. geophones, accelerometers, and piezoelectric sensors), to measure the vibrations of the structure (Rickenmann, 2017a). Seismometers are typically installed on streambanks and primarily record seismic waves generated by the transport of larger particles and the turbulent water flow (Dietze et al., 2019).

Many studies summarize successful investigations with the impact plate systems in the field and within the frame of controlled flume experiments (e.g. Bogen and Møen, 2003; Krein et al., 2008; Tsakiris et al., 2014; Mao et al., 2016; Wyss et

al., 2016b, c; Kuhnle et al., 2017). Among these systems, the Swiss plate geophone (SPG) system was installed and tested in more than 20 steep gravel-bed streams and rivers, mostly in the European Alps (Rickenmann, 2017a; Nicollier et al., 2021, 2022b). For the SPG, linear and power-law calibration relationships were developed between measured signal properties and bedload transport rate or mass (Rickenmann et al., 2014, 2020; Rickenmann and Fritschi, 2017; Habersack et al., 2017; Wyss et al. 2016a; Kreisler et al., 2017; Kuhnle et al., 2017; Nicollier et al., 2021, 2022b; Coviello et al., 2022). There are

similarities between calibration realtionships based on bedload samples at various field sites, but it is not well understood why the linear calibration coefficients for total mass flux can vary by about a factor of 20 among individual samples from different sites, or by about a factor of six among the mean values from different sites (Rickenmann et al., 2014; Rickenmann and Fritschi, 2017). Impact tests and controlled flume experiments allowed to identify the grain size distribution as one reason for the variability of the signal response (Nicollier et al., 2021). Another reason explaining a part of the variability is

the lateral signal propagation that occurs when medium-sized to larger particles impact on different locations of a



neighbouring plate, or on the nearby concrete bed (Antoniazza et al., 2020; Nicollier et al., 2022a; Chen et al., 2022a). Furthermore, both field (Rickenmann et al., 2014) and flume (Wyss et al. 2016b; Kuhnle et al., 2017; Nicollier et al., 2021) observations showed that the flow velocity plays also an important role for the impulse-based calibration factor (see also Nicollier et al., 2022a, 2022b).

Most calibration relations for impact-type acoustic monitoring techniques such as the SPG were based on an analysis of the signal in the time domain (e.g. by impulse counts above a threshold level excluding the system noise). Several studies with controlled flume experiments using such systems showed that the signal amplitude contains information about the grain size of the transported bedload, and enables to detect particles as small as about 5 to 10 mm (Beylich and Laute, 2014; Mao et al., 2016; Wyss et al., 2016a). Some studies further indicated indicate that the size of the transported bedload particles can also

be related to the frequency content of the signal registered by acoustic sensors (Bogen and Møen, 2003; Barrière et al., 2015; Wyss et al., 2016 b; Rickenmann, 2017a). A combination of both amplitude and characteristic frequency represents potentially a more robust identification of the transported particle sizes (Barrière et al., 2015; Wyss et al., 2016 b; Nicollier et al., 2022a, 2022b).

The goal of this study is to compare the performance of four surrogate acoustic measuring systems for bedload transport in

reference to direct bedload measurements. These four acoustic measuring systems are: (i) the SPG, on which many studies of the Swiss Federal Research Institute (WSL) and other research groups focused in the past twenty years, (ii) the miniplate accelerometer (MPA), a smaller variant of the SPG system developed by WSL, (iii) the GP-Acc (for geophone plate accelerometer) which uses the same frame as the SPG system but equipped with an accelerometer sensor instead of a geophone sensor, and (iv) the Japanese pipe microphone (JPM), a system which has been installed in many mountain

streams in Japan. An illustration of the four measuring systems is presented in Figure 1, showing the devices as installed at the Erlenbach stream. Calibration measurements for the surrogate systems were obtained at four locations, namely at three field sites in the Swiss Alps: the streams Erlenbach, Avançon de Nant, and the Albula river, and at an outdoor flume facility in Obernach (Germany), allowing for controlled flow conditions and sediment feeding during the experiments. In this paper, we discuss the temporal stability of the signal (noise) under non-bedload transport conditions, the quality of the calibration

relationships obtained from the direct bedload measurements, and we identify some possible reasons for the poorer performance of the MPA as compared to the other investigated systems.


Earth **Surface**
**Dynamics**
Discussions

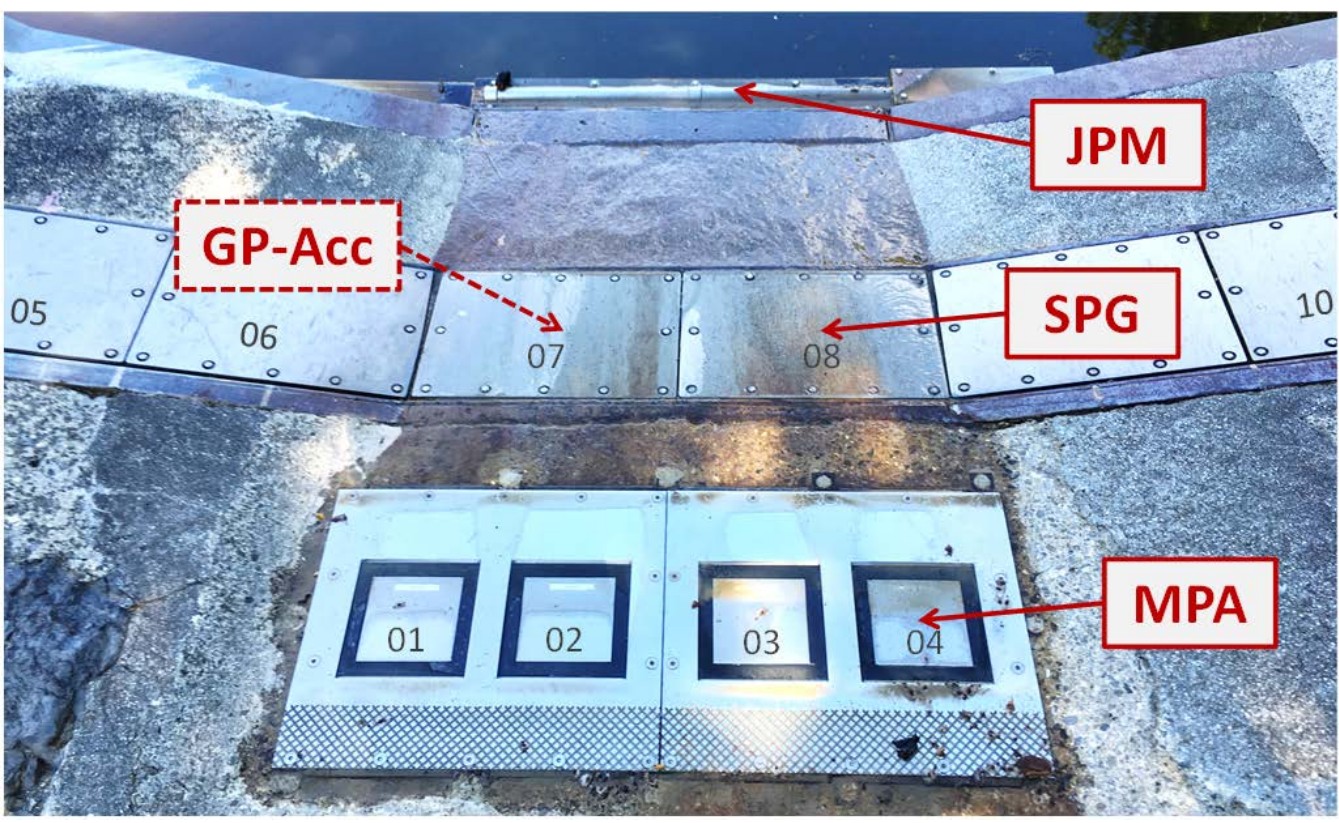

**Figure 1: The four acoustic measuring systems considered in this study: (i) the Swiss plate geophone (SPG), (ii) a variant of the SPG system, where the steel plate is equipped with an accelerometer sensor instead of a geophone sensor (GP-Acc), (iii) the miniplate accelerometer (MPA), and (iv) the Japanese pipe microphone (JPM). The set-up in this picture is the one at the Erlenbach stream in Switzerland (with flow direction into the retention basin from bottom to top). Number in dark grey refer to plate numbers.**

## 2 Surrogate measuring systems and direct bedload transport measurements

## 2.1 Surrogate measuring systems

The four surrogate measuring systems SPG, MPA, GP-Acc, and JPM were deployed in this study. The first three systems were developed at our research institute WSL, whereas the JPM was purchased from Japan. The major dimensions of the impact systems are depicted in Figure 2.





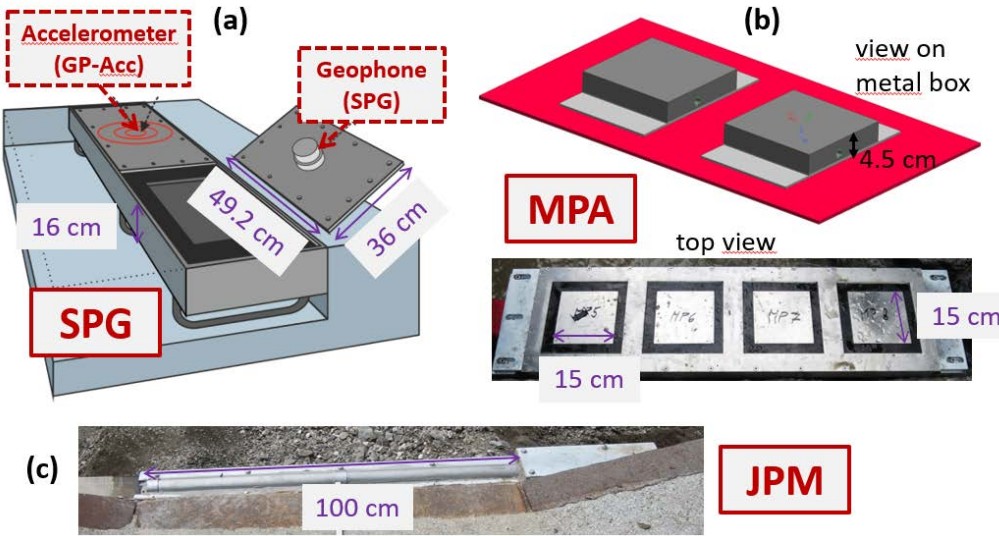

**Figure 2. Sketches with the main dimensions of the four systems. For (a) the SPG and (b) the MPA, the plate thickness is 15 mm and 8 mm, respectively. For (c) the JPM, the pipe diameter is 48 mm, and its wall thickness 3 mm.**


The SPG system consists of a steel plate mounted flush with the streambed and equipped with a geophone sensor fixed from underneath in the centre of the plate. The steel plate has standard dimensions of $L \times B \times T = 0.360$ m $\times 0.492$ m $\times 0.015$ m, where L is the downstream length, B is the transversal width, and T is the thickness of the steel plate. A 20DX geophone from Geospace Technologies (Houston, Texas, USA) in a PC801 LPC Land-case is fixed in a water-tight aluminium case to

the bottom of the steel plate. The geophone sensor measures impact shocks generated by bedload particles moving over and impacting the plate. The sensor contains a magnet in a coil as inductive element. The relative movement between the coil and the magnet induces a current proportional to the velocity of the impacted plate (Rickenmann et al. 2012). The output range of the sensor is +/- 10 V (Table 1). Each plate covers a unit stream width of 0.5 m, and multiple steel plates are often mounted side-by-side into a steel canal frame, one segment typically covering 2.5 to 3 m of stream width. The plates are

acoustically isolated from each other and from the steel canal by elastomer elements in which they are embedded, to minimize the recording of extraneous vibrations (e.g. from particle impacts on neighbouring plates, or on the concrete up- or downstream of the steel canal).

**Table 1: Sensor types used in the measurement systems in this study, and their respective output range.**

| Sensor | Output range | Used in measuring system | Full name of measuring system |
|---|---|---|---|
| Geophone | +/- 10 V | SPG | Swiss Plate Geophone |
| Accelerometer | +/- 500 g | MPA, GP-Acc | MiniPlate Accelerometer; Geophone Plate Accelerometer |
| Microphone | +/- 5 V | JPM | Japanese Pipe Microphone |




The GP-Acc is a variant of the Swiss plate geophone system, using the same steel frame set-up but with an accelerometer instead of a geophone sensor to measure the vibrations of the plate. A general-purpose accelerometer model KS78.10 from Metra Mess- und Frequenztechnik (Radebeul, Germany) was installed, with an output range of +/- 500 g (Table 1). Calibration measurements with the GP-Acc were made in two slightly different set-ups: at the Erlenbach stream, the accelerometer sensor was mounted next to the geophone sensor under the same impact plate, and at the Albula stream every second plate was equipped with either a geophone or an accelerometer sensor (see also Table 2).

The MPA system was developed with the idea of constructing a more compact system than the SPG. In addition, it may be expected that the MPA and GP-Acc systems can better detect smaller particles than the SPG system, since an accelerometer is designed to better pick up higher frequencies (above about 1 kHz) than a geophone sensor. The core unit of the MPA system is a metal box that houses an accelerometer sensor, mounted at the underside and in the centre of a compact steel box. The bottom part of the box is closed by a thin steel plate of somewhat larger dimensions than the top surface with dimensions of L × B × T = 0.150 m × 0.150 m × 0.008 m (Fig. 2). The entire metal structure is embedded in elastomer layers, and partly covered by a metal frame to be robust against forces generated by the water flow and bedload-particle transport. The sensor used is the miniature adhesive mount IEPE accelerometer model 805M1 from Measurement Specialties (Aliso Viejo, CA, USA), with an output range of +/- 500 g (Table 1). The MPA system is somewhat similar in terms of plate dimension and sensor type to the impact plate system used primarily in UK studies (Downs et al., 2016; Raven et al., 2009, 2010; Reid et al., 2007; Richardson et al., 2003), although this latter system was fixed more rigidly to the streambed.

Finally, the Japanese pipe microphone (JPM) is a system that was developed in Japan by Hydrotech Company (Kouzukeda, Shiga, Japan). At the various Japanese field sites, it is typically placed transversally to the flow direction across the streambed, and buried by roughly half its diameter in cement at a stable bed section such as a check dam or a sill (Mizuyama et al. 2010a, 2010b). For the installation at the Erlenbach stream, we embedded the pipe in elastomer layers, so that about 40 % of the upper pipe surface was exposed to the flow. In this way, we wanted to guarantee a stable configuration of the exposed pipe surface, not influenced by a possibly degrading cement layer over time. The steel pipe is filled with air, and a microphone records the sound (pressure waves) generated by particles impacting the pipe. In most studies that used the JPM, the raw signal is first treated with a band pass filter, and a wave detector determines the envelope of the signal. Based on the envelope, the number of waves or wave peaks exceeding the threshold level is used to derive the pulse counts (Tsutsumi et al., 2018). Note that the pulse counting is very similar to the counting of signal packets as described in studies with the SPG system; a packet is defined as a continuous time section of the signal corresponding to one impact of a bedload particle (Wyss et al., 2016a; Nicollier et al., 2022b). For the observations presented in this study, we recorded the raw signal of the microphone and analyzed it in the same way as for the other three systems (see section 2.2. below). The output range of the microphone is +/- 5 V (Table 1).

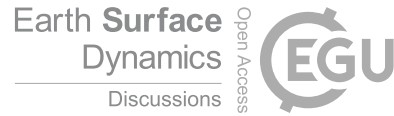

## 2.2 Signal recording and processing of calibration measurements

During a calibration measurement, i.e. the time period of direct bedload sampling, the full raw signal is recorded for each measuring system (Table 2). For the SPG system at all sites and for the GP-Acc at Albula and for the MPA at Avançon de

Nant and the Obernach flume, a recording frequency of 10 kHz was used. For the non-SPG systems at Erlenbach, higher recording frequencies of first 50 kHz and later 20 kHz were used (Table 2). In these latter cases, the raw signal was down-sampled to 10 kHz before further processing for the calibration analysis, to avoid any possible bias due to differences in sampling frequency when comparing the MPA measurements from the Erlenbach and the other sites.

**Table 2: Main characteristics of raw data acquisition and processing for the different measuring systems. The listed threshold value $A_{min}$ was used for the processing of the raw data. The $A_{min}$ value in square brackets refers to the threshold value used for impulse counts for the continuous storage of the summary values in minute intervals; if no square bracket is present, the listed $A_{min}$ value was used for the calculation of the summary values. Down-sampling of the original sampled signal was made for the analysis of this study for better comparability between measuring systems and calibration locations.**

| Site | Measuring period | Sensor type | $A_{min}$ | Units | Sampling frequency (kHz) | Down-sampling (10 kHz) |
|---|---|---|---|---|---|---|
| Erlenbach | 05.07.2013 – 16.05.2014 | JPM | 0.1  [0.05] | V | 50 | yes |
| | 16.05.2014 – 19.06.2016 | GP-Acc | 2.5  [0.1] | V | 50 | yes |
| | | JPM | 0.1  [0.05] | V | 50 | yes |
| | 24.06.2016 – 31.12.2020 | MPA | 2.5  [2.0] | g | 20 | yes |
| | | GP-Acc | 2.5  [2.0] | g | 20 | yes |
| | | JPM | 0.1 | V | 20 | yes |
| | 05.07.2013 – 31.12.2020 | SPG | 0.1 | V | 10 | |
| Albula | 16.04.2015 – 23.08.2016 | GP-Acc | 2.5  [0.4] | g | 10 | |
| | 24.08.2016 – 31.12.2020 | | 2.5 | g | 10 | |
| | 16.04.2015 – 31.12.2020 | SPG | 0.1 | V | 10 | |
| Avançon | 09.09.2015 – 31.12.2020 | MPA | 2.5 | g | 10 | |
| | 09.09.2015 – 31.12.2020 | SPG | 0.1 | V | 10 | |
| Obernach | 2018, 2019, 2020 | MPA | 2.5 | g | 10 | |
| | | SPG | 0.1 | V | 10 | |




During normal flow monitoring conditions (including the periods with direct bedload sampling), a pre-processing of the vibration signal provides summary values; due to data storage limitations, the full raw signal is not always recorded. For all

four measuring systems, the following summary values were recorded for each one minute time interval (Rickenmann et al. 2014): (i) whenever the voltage or acceleration exceeds a pre-selected threshold value Amin in the positive domain, counted as an impulse and the summed impulse counts (*IMP*) are stored; (ii) the maximum value of the signal amplitude per one second interval is determined and summed over the one minute recording interval; (iii) the root mean square of the time-varying signal is calculated for each second, then squared and summed over one-minute intervals, to represent the sum of the

squared amplitudes values (*IQA*).

The threshold amplitude $A_{min}$ (Table 2) was first defined for the SPG system with the aim that this value should be clearly above a mean noise level. The threshold values $A_{min}$, initially selected for the other systems, were slightly modified over time, so that the registered *IMP* values and the selected $A_{min}$ value better scaled with the SPG measurements. Previous studies with the SPG system showed that at many sites a linear calibration relation between the *IMP* and the bedload mass *M*

transported over the plates provides a good description of the calibration measurements, where $k_b$ is the linear calibration coefficient (e.g. Rickenmann et al., 2012, 2014; Nicollier et al., 2021):

$$IMP = k_b M \tag{1}$$

To check, how well a linear relation approximates the calibration data, and for comparison with earlier studies with the SPG system, we also determined a power-law relation for the calibration measurements, with the empirical coefficient α and

exponent *β*:

$$IMP = \alpha M^{\beta} \tag{2}$$

Using these *IQA* values for those minute intervals when the *IMP* values are zero, i.e. for time steps with no or negligible bedload transport activity, an average noise level of the signal was calculated as square root of (*IQA*/60).

Examples of the signal output for the SPG system can be found in previous papers (Chen et al., 2022a; Nicollier et al.,

2022a, 2022b; Rickenmann et al., 2012, 2014; Wyss et al., 2016a). For the MPA measurements at the Erlenbach, the effect of down-sampling of the raw signal from 20 kHz to 10 kHz was checked with the help of the so-called packet data (storage of the raw signal only for the time periods when a packet is detected) that was available for the years 2016 to 2020 with a time resolution of 20 kHz. It was found that maximum amplitude is practically not affected by the down-sampling (Fig. S1), whereas the centroid frequency shows much higher values for the original 20 kHz data (in the range of about 3.5 to 8.5 kHz)

than for the down-sampled 10 kHz data (in the range of about 1.0 to 3.0 kHz) (Fig. S2), which is not surprising given that frequencies can only be determined without aliasing for values lower than half the sampling frequency (Nyquist frequency, e.g. Onajite, 2014). As a result, the number of impulses (*IMP*) are also larger for the original 20 kHz data than for the down-





sampled 10 kHz data, particularly at the higher transport rates (Fig. S3). This result confirms the need for using the same
sampling frequency at all sites and time periods for the purpose of our study.

Comparing the signal response between the SPG and the MPA systems per unit stream width, one has to account that the
ratio $r_A$ of the respective plate surface areas $A$ is $r_A = A_{MPA}/A_{SPG} = (4{\times}0.15)$ m$^2$ / $(2{\times}0.492{\times}0.358)$ m$^2$ = 0. 09m$^2$ / 0.352 m$^2$ =
0.256, and the ratio of the number of sensors $r_S$ is 4/2 = 2 (with all numbers applying for 1 m width). For an equal density of
particles impacting the plate per unit surface area, and assuming a similar signal response, one could roughly expect that the
MPA system should record about half as much impulses *IMP* as the SPG system for a given width, e.g. over 1 m at the
Erlenbach or at Obernach.

**2.3 Field and flume sites with direct bedload measurements**

The Swiss field sites and the Obernach outdoor flume facility were already described in some detail in other publications
(e.g. Nicollier et al., 2022b; Rickenmann et al., 2012; Antoniazza et al., 2022). Therefore, only a brief overview is provided
here. Table 3 summarizes the channel and flow characteristics and the year of calibration measurements. Table 4 lists the
sampling method and the plate numbers of the surrogate systems used at different sites. The number of calibration
measurements and the total sampled mass with the direct measurement methods at the different sites and for different
measurement systems is illustrated in Figure 3. For most sites and systems, the number of single calibration measurements
was typically between about 30 and 80, and the total sampled bedload mass typically varied between about 1500 and 3000
kg.


**Table 3: Channel and flow characteristics from in situ measurements made during the calibration campaigns at the three field
sites and at the Obernach flume. The years of the field calibration campaigns, are also indicated.**

| Field or flume site | Location | Bed slope [%] [a] | Mean flow velocity $V_w$ [m/s] [b] | Channel width [m] | Year of calibration data used | More site-specific details [c] |
|---|---|---|---|---|---|---|
| Erlenbach | Alpthal (CH) | 16 | 5 | 3 | 2013 - 2020 | [1], [2], [3], [5] |
| Albula | Tiefencastel (CH) | 0.7 | 2.6 | 15 | 2018 | [3], [4], [5], [6] |
| Avançon de Nant | Plans-sur-Bex (CH) | 4 | 1.3 | 5 | 2019/2020 | [5], [7] |
| Obernach flume | Obernach (DE) | 4 / 0.7 | 1.6 / 2.4 / 3.0 | 1 | 2018/2019/ 2020/(2021) | [3], [4], [5], [8], [9] |

[a] Gradient measured upstream of the SPG plates

[b] Depth-averaged mean flow velocities measured during the calibration measurements at field sites; at Obernach measured 0.1 m above
SPG





(c) [1] Rickenmann et al. (2012), [2] Wyss et al. (2016c), [3] Nicollier et al. (2021), [4] Nicollier et al. (2022a), [5] Nicollier et al. (2022b), [6] Rickenmann et al. (2020), [7] Antoniazza et al. (2022), [8] Chen et al. (2022a), [9] Chen et al. (2022b)

**Table 4: Bedload sampling characteristics for calibration measurements at the different field and flume sites. Plate numbers are indicated by 2 digits. Those separated by a comma indicate that all plates recorded particle transport entering the sampler. Plate numbers on the same line experienced the same bedload transport during a given calibration measurement. Numbers are listed in orographic order from left to right (in flow direction). For the JPM, calibration measurements are only available at the Erlenbach site ("yes"). A horizontal dash ("-") indicates that no measurements are available for a given system or set-up.**

| Field or flume site | Sampling technique | Sampling width | SPG plate(s) | MPA plate(s) | GP-Acc plate(s) | JPM |
|---|---|---|---|---|---|---|
| Erlenbach | automatic basket sampler | 1 m | 07, 08 | 01, 02, 03, 04 | 07, 08 | yes |
| Albula | crane-mounted net sampler | 0.5 m | 06 | - | 05 | - |
| Avançon | manual basket sampler | 0.5 m | 05 | 08, 07 | - | - |
| | | | 06 | 06, 05 | | |
| | | | 07 | - | | |
| | | | 08 | 04, 03 | | |
| Obernach | manual sediment feed | 1 m | 02, 01 | 04, 03, 02, 01 | - | - |

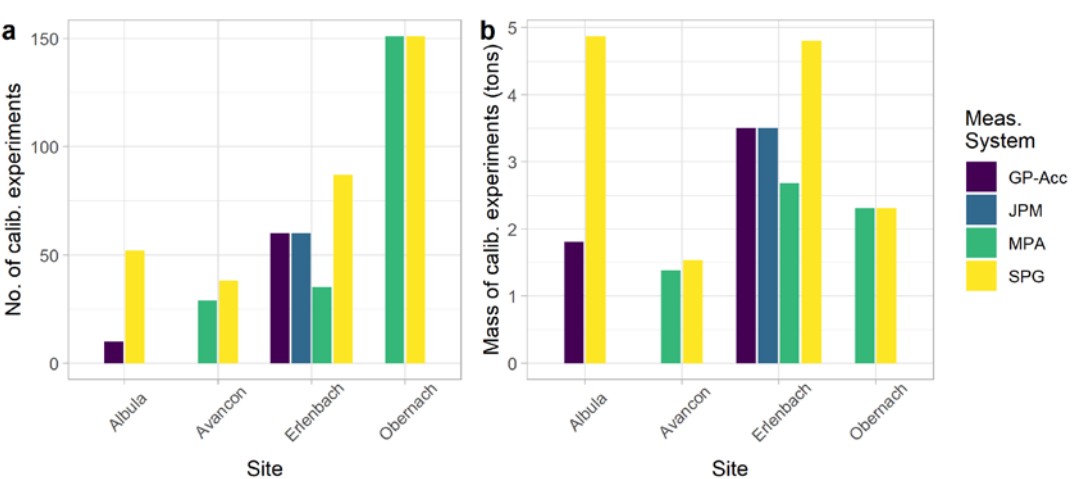

**Figure 3: Amount of calibration data in terms of (a) number and (b) total sampled mass with direct measurement methods at the different sites and for different measurement systems. At the Avançon, some calibration experiments included very high transport rates of fine grains (potential shield or cover effect on the plates), and those were removed for further analysis (Nicollier et al., 2022b; Antoniazza et al., 2022).**






# 3 Results

## 3.1 Noise level during non-transport periods

The variability of the noise level for each year is illustrated in Figure 4 for the four measuring systems and at all four measuring sites. For the SPG and MPA systems at all sites and for the GP-Acc at the Albula river, the noise level remains

fairly constant over the (maximum) seven years of observations, and most of the time the noise level is clearly below the threshold level $A_{min}$ for impulse counts. However, at the Erlenbach site, a clear increase in noise level is observed for the GP-Acc and the JPM starting in the year 2016 (Fig. 4c, 4d). Particularly for the JPM system, the noise level often exceeded the threshold level $A_{min}$ during this more recent period, thus also biasing the impulse counts for this system. The reason for this increase in noise level is likely due to a change in the data acquisition system for the GP-Acc and the JPM at the Erlenbach

in June 2016 (Fig. S4). Previous to this date, the signal was recorded by a PCI data acquisition card embedded in the computer, whereas after this date the same signal was recorded via Ethernet cable and separate acquisition modules. This temporal instability of the noise level has to be accounted for when assessing the calibration relationships for different time periods. At the Albula site, an unstable behaviour of some of the accelerometer sensors (GP-Acc) was very pronounced; the signal of these sensors was hardly usable at all, and these sensors had to be disconnected from the recording system.


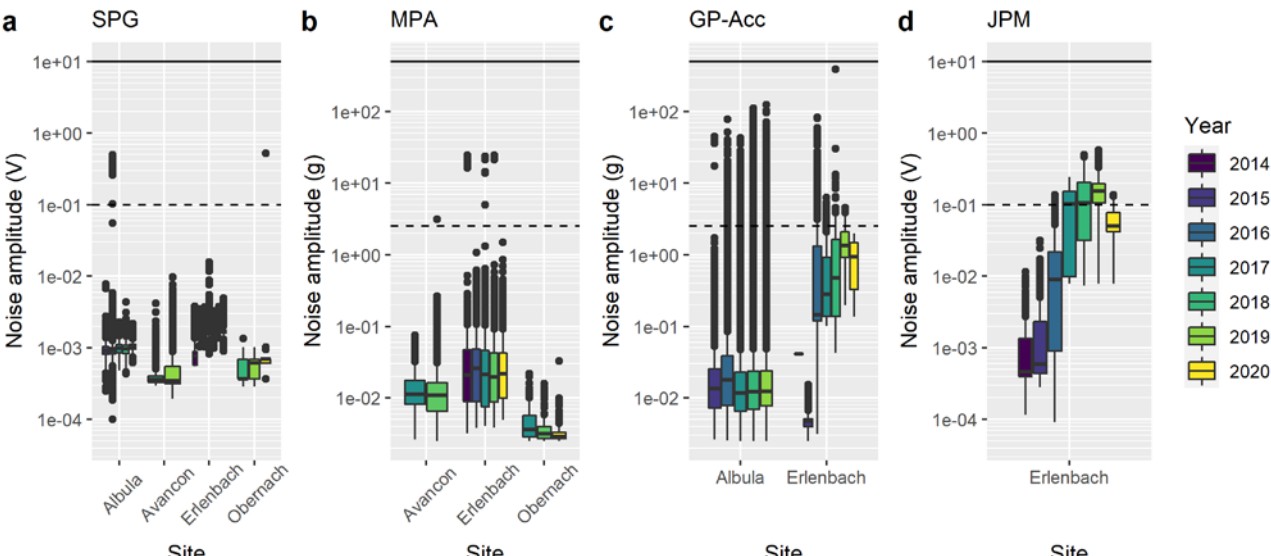

**Figure 4: Background noise level of SPG (a), MPA (b) GP-Acc (c) and JPM (d) for each year at different sites during non-transport conditions. For the Obernach site, the noise levels were determined as the median absolute amplitudes of the raw signal of each experiment (since no IQA values were recorded as for the field sites). The dashed and solid lines indicate the threshold for impulse counts ($A_{min}$) and the upper limit of output range, respectively.**



## 3.2 Calibration relationships for different systems and sites

We first compare the calibration of different measuring systems for the Erlenbach site, because it is the site with the largest number of systems and the longest observation periods (Fig. 5, Table 2). While the number of impulses per bedload mass $k_b$
is relatively stable and stays constant over time for the SPG system, we see an increased variability for the other measurement systems (Fig. 5a). The GP-Acc and the JPM systems also show a good performance for the period before June 2016, but for the later period the calibration data show a much-increased scatter around the mean calibration relation based on observations before June 2016 (Fig. 5b).

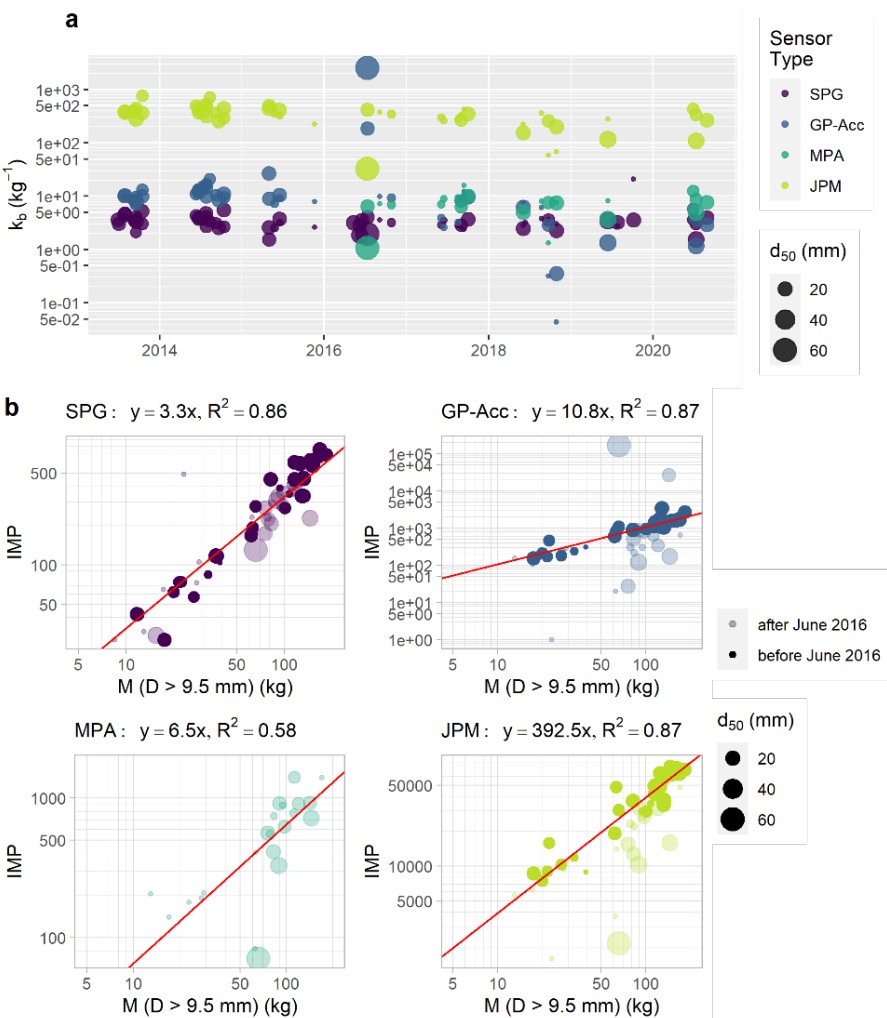

**Figure 5: Calibration measurements at the Erlenbach. (a) Temporal evolution of ratio of impulses per transported mass $k_b$. (b) Calibration relation for each measurement system, showing the number of impulses *IMP* vs. bedload mass *M*. Samples lighter than 5 kg were excluded, and only grains with *D* > 9.5 mm were retained from the direct samples. In June 2016, the data acquisition system recording the signal of the GP-Acc and the JPM systems was changed; the same acquisition system was used to record the MPA**





**measurements (starting in 2016). For the GP-Acc and the JPM, regression equations refer only to the period before June 2016. The SPG measurements were recorded by a different system, which did not change in the period 2013 to 2020.**


We now compare the calibration relations for the SPG and MPA systems at the Erlenbach site, the Avançon de Nant site, and the Obernach flume, to consider also the between-site variability. For the SPG system, we find fairly good linear calibration relationships for all three sites, the squared correlation coefficient values $R^2$ varying between 0.78 and 0.86 (Fig. 6a). Similar calibration relations were reported by Nicollier et al. (2021, 2022b) in terms of *IMP* or signal "packet" counts

vs. unit bedload transport rate. A packet is defined as a continuous time section of the SPG signal corresponding to one impact of a bedload particle. For the MPA system, we find weaker linear calibration relationships for Erlenbach and Avançon de Nant site, with $R^2$ values of 0.54 and 0.58, while at the Obernach site there is only a very poor correlation between *IMP* and *M* (Fig. 6b). The smallest $R^2$ value obtained for this site may be partly due to preferential particle trajectories (along the sidewalls) caused by the fixed rough bed upstream, and due the smaller total impact surface of the

MPA as compared to the SPG.

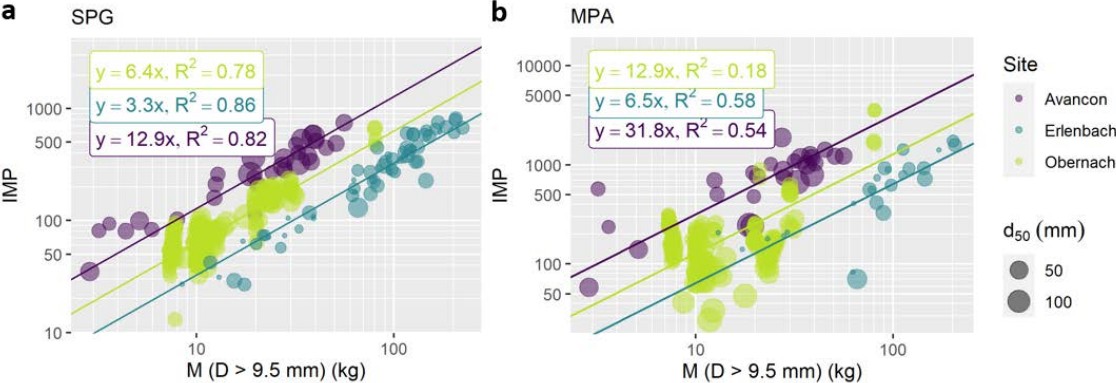

**Figure 6: SPG impulses *IMP* vs. total transported mass *M* (D > 9.5mm) for the calibration measurements at different sites. Samples lighter than 5 kg were excluded for the calculation of the regression for the Avançon. Results are (a) for the SPG, (b) for the MPA.**

For a given bedload mass *M*, the number of impulses *IMP* increases for both the SPG and MPA from the Erlenbach site to the Obernach flume to the Avançon de Nant site (Fig. 6), which is reflected in an increasing linear calibration coefficient $k_b$.

We have evidence for the SPG system that this increase in signal response is due to a decrease in mean flow velocity (Erlenbach > Obernach > Avançon) and due to a smoother channel roughness at the Erlenbach compared to Obernach and Avançon (Nicollier et al., 2022). These two factors both influence the transport mode of the particles and the signal response. We suspect that these reasons also apply to the responses of the MPA system, which is qualitatively and quantitatively similar between the different sites.



A comparison of the calibration relationships obtained for the different sites and the different measuring systems is compiled in Table 5 and illustrated in Figure 7. For all four sites (and if for the Erlenbach only periods with a stable noise level of the signal are considered), good correlations with $R^2$ values > 0.77 were obtained for the SPG, GP-Acc, and JPM systems (Fig. 7a). For the MPA system, $R^2$ values are moderately good at the Avançon de Nant and Erlenbach sites (0.54 and 0.58, respectively) but very poor (0.18) at the Obernach site (Fig. 7a). Concerning the inter-site variability in terms of the $k_b$

values, the relative differences appear to be similar at different sites when comparing both GP-Acc with SPG and MPA with SPG (Fig. 7b).

**Table 5. Linear and power-law calibration relationships between *IMP* and *M* obtained for the different sites and the different measuring systems. *N* is the number of samples used for the regression calculation. [*] determined for the period before June 2016.**

| Site | System | $N$ | Linear relation | $R^2$ | Power-law relation | | $R^2$ |
|---|---|---|---|---|---|---|---|
| | | | $IMP = k_b M$ | | $IMP = \alpha M^{\beta}$ | | |
| | | | $k_b$ | | $\alpha$ | $\beta$ | |
| Erlenbach | SPG | 87 | 3.3 | 0.86 | 2.6 | 1.06 | 0.87 |
| | MPA | 35 | 6.5 | 0.58 | 8.4 | 0.94 | 0.58 |
| | GP-Acc[*] | 60 | 10.8 | 0.87 | 7.6 | 1.08 | 0.88 |
| | JPM[*] | 60 | 392.5 | 0.87 | 530.3 | 0.93 | 0.88 |
| Albula | SPG | 52 | 12.9 | 0.91 | 13.8 | 0.98 | 0.91 |
| | GP-Acc | 10 | 21.5 | 0.87 | 49.7 | 0.81 | 0.92 |
| Avançon | SPG | 38 | 12.9 | 0.82 | 23.2 | 0.84 | 0.85 |
| | MPA | 29 | 31.8 | 0.54 | 86.6 | 0.74 | 0.62 |
| Obernach | SPG | 151 | 6.4 | 0.78 | 6.5 | 0.99 | 0.78 |
| | MPA | 151 | 12.9 | 0.18 | 30.7 | 0.66 | 0.25 |


   Comparing the performance of the MPA and SPG only, a generally (clearly) lower quality of the calibration relations is observed at all sites for the MPA (Fig. 5, 6, 7; Table 5). This could be due in part to generally weaker signal responses of the MPA system triggered by larger particle impacts, as is evidenced from the observations at the Obernach flume site, for which *IMP* values as a function of *M* are plotted separately for different grain size classes in Figure 8. For total bedload

masses *M* in the range 4 kg < *M* < 50 kg, there is a variability of the signal response ($k_b$ value) of up to about a factor of 10 for the SPG system, without a clear stratification with grain size class. In contrast, for the MPA system the variability of the signal response is more than a factor of 10 with larger particles clearly generating less impulses for a given bedload mass. The comparison in Fig. 8 also shows that the MPA is more sensitive to smaller grains (ca. 20-50 mm), whereas the SPG is more sensitive to larger grains (ca. 50-100 mm). Given the greater proportion of smaller grains as compared to bigger ones in

natural streams, it may explain the generally larger $k_b$ values for the MPA than for the SPG at a given site (Table 5).





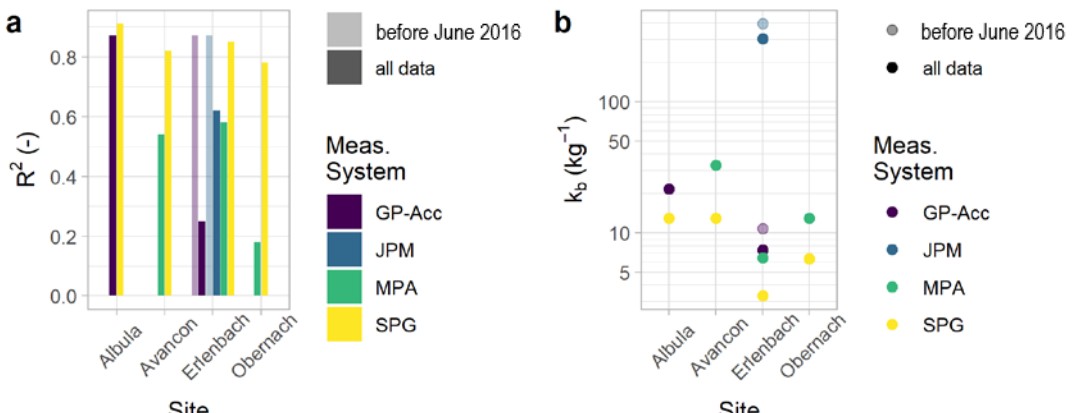

**Figure 7: Overview of calibration relations for different measurement systems at different sites. (a) Correlation coefficient R² for the linear regression, (b) linear calibration coefficient $k_b$.**

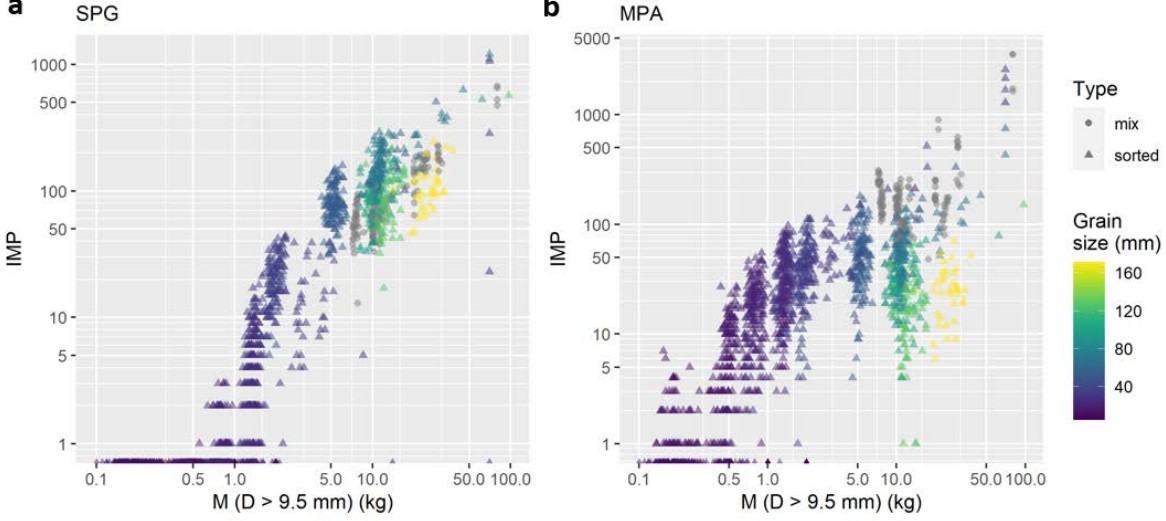

**Figure 8: Obernach flume site. (a) SPG impulses and (b) MPA impulses vs. total transported mass *M*, for both flume experiments with single grain size classes and those with mixed grain sizes. The size class indicates the median grain size present in each experiment.**

## 4 Discussion

### 4.1 Amplitude and frequency response to particle impact for the SPG and MPA systems

The generally relatively weaker signal response of the MPA system to the impact of larger particles as compared to the SPG system (Fig. 8) is likely due to two factors observed for the Obernach flume experiments. First, there is generally a larger (relative) variability in the maximum amplitude response over all the experiments for a given grain size class for the MPA





system (Fig. 9). Second, the tendency for increasing maximum amplitude with increasing grain size is only observed for $D$

up to about 76 mm for the MPA system, whereas for larger particles the mean maximum amplitude even decreases (for the

sorted experiments) or remains approximately constant (for mixture experiments) (Fig. 9). Two hypotheses may be raised to

explain this behaviour: First, the metal plate (box) of the MPA system is embedded in several layers of elastomer,

representing a much larger volumetric proportion of elastomer vs. steel. As such, the metal plate is mounted in a softer, more

deformable environment with higher absorption capacity than for the configuration of the SPG system. Second, the ratio $r_A$

of the total plate surface area of the MPA to that of the SPG per 1 m channel width is about 0.26 (section 2.2), resulting in a

greater probability for the MPA system that large particles (that were limited in number) impacted onto the area between the

plates or on the edges of the plates, than was the case for the SPG system. (In fact, the number of particles per experimental

run decreased towards larger grain sizes.) In contrast to the differing amplitude response of the two systems, the centroid

frequency response for changing grain sizes was rather similar for the MPA and the SPG system (Fig. 10). For both systems,

the gradient of decreasing frequency is somewhat steeper in the larger particle data domain and weaker in the smaller particle

domain.

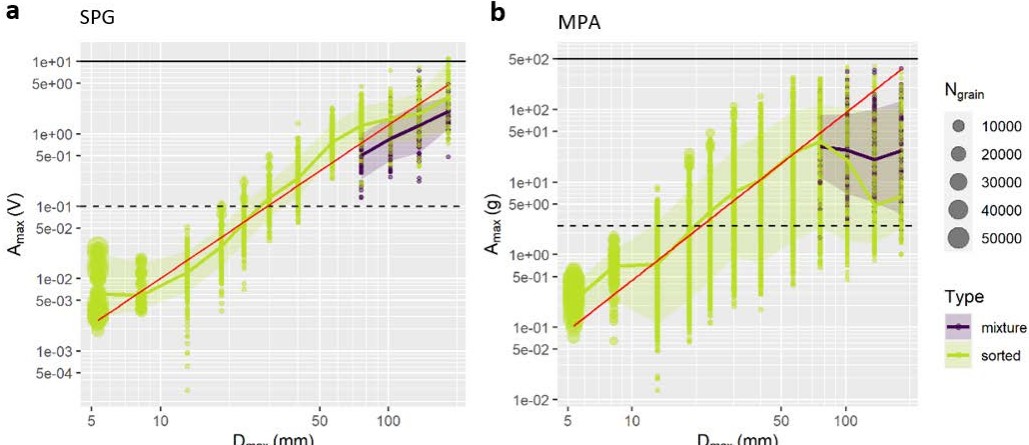

**Figure 9: Obernach flume site. Maximum (positive) amplitude $A_{max}$ recorded per experimental run vs. grain size $D$. The power law regressions are based on the sorted (single grain size class) experiments only. (a) For the SPG system, (b) for the MPA system**





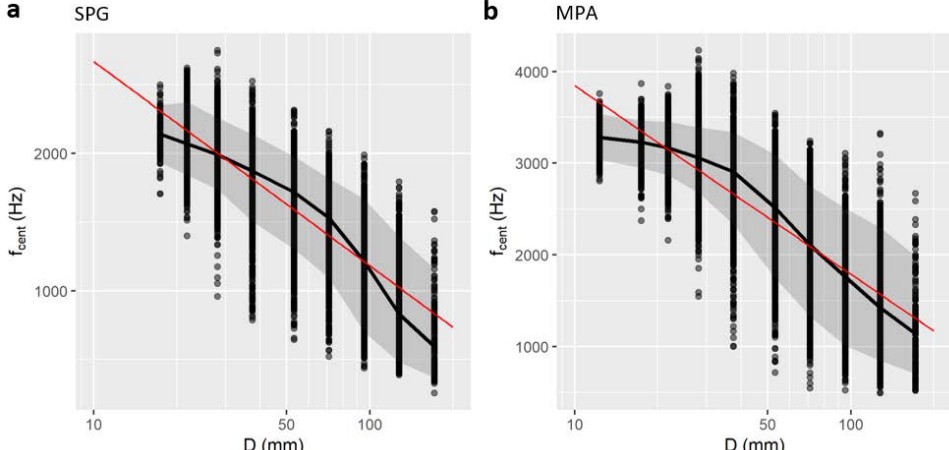

**Figure 10: Obernach flume site. Centroid frequency $f_{cent}$ vs. grain size $D$ for packets with the 5 largest amplitudes per experimental run vs. grain size $D$. Only single grain size class experiments were used. (a) For the SPG system, (b) for the MPA system**

### 4.2 Velocity effect on signal response of the SPG and MPA systems

The effect of changing flow velocities (and thus particle velocities or transport mode) was investigated with the calibration measurements from the Obernach flume site, showing that that different flow velocities resulted in a larger scatter around a mean calibration line for the MPA system than for the SPG system (Fig. 11). Considering the velocity effect on the amplitude response of the two systems, we observed that the variability of the amplitude for a given grain size (class) is clearly larger for the MPA than for the SPG system (Fig. 12). The maximum amplitude of a signal packet correlates with the number of recorded impulses, which may be partly responsible for the larger scatter of the data around a mean calibration relation (as in Fig. 11) and the generally weaker associated correlation coefficient for the MPA as compared to the SPG system (Table 5). Regarding the effect of flow velocity on the centroid frequency, its variability was similar for both systems for $D > 40$ mm, whereas for $D < 40$ mm a changing flow velocity leads to a more variable centroid frequency for the SPG than for the MPA system (Fig. S5). If frequency information was used for grain size classification, the MPA system (or using accelerometer sensors in general) would have a potential advantage to better classify smaller particles compared to the SPG system (or using geophone sensors).



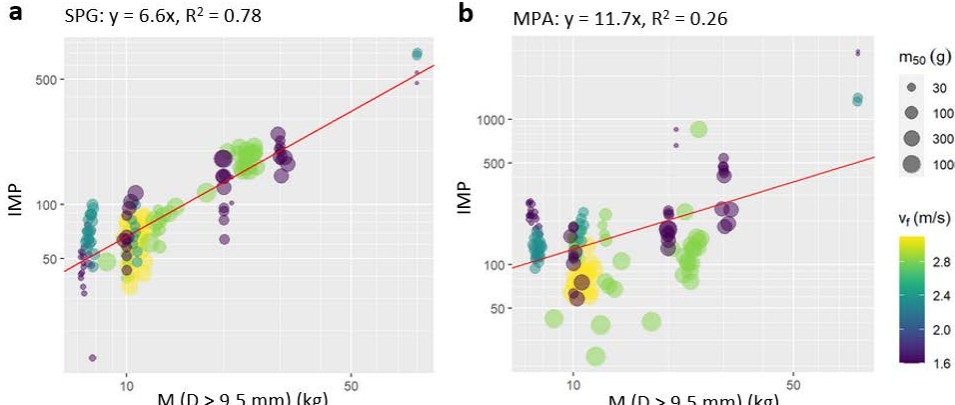

**Figure 11: Obernach flume site. Signal impulses *IMP* vs. total transported mass *M*, for flume experiments with bedload material of mixed grain sizes. $m_{50}$ is the median total particle mass, $v_f$ is the water velocity 10 cm above the impact plates. (a) For the SPG system, (b) for the MPA system**

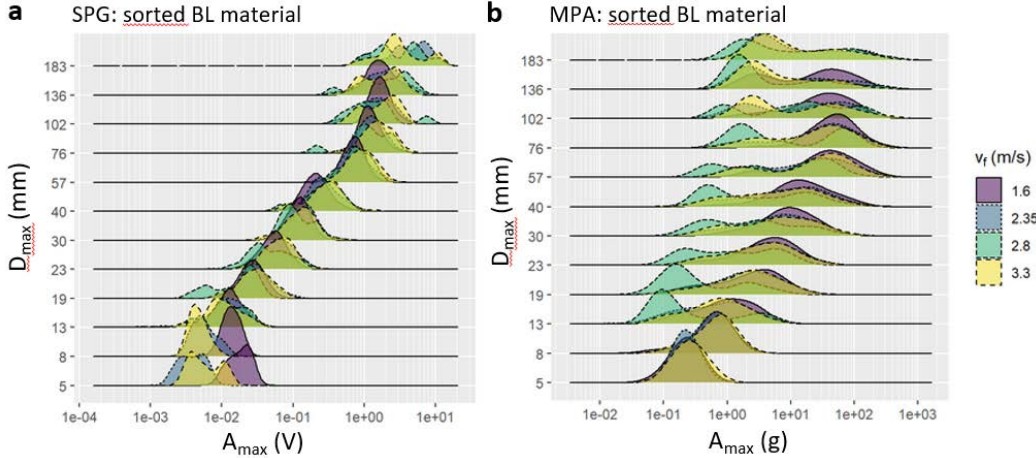

**Figure 12: Obernach flume site. Maximum grain size $D_{max}$ vs. maximum (positive) amplitude $A_{max}$ for the sorted grain size experiments. $v_f$ is the water velocity 10 cm above the impact plates. (a) For the SPG system, (b) for the MPA system. $A_{max}$ represents the largest amplitude of the 2 geophone sensors (a) or of the 4 accelerometer sensors (b). $D_{max}$ represents a value 20% larger than the characteristic diameter of a given grain size class.**


## 4.3 Noise level and signal "saturation" for frequent particle impacts

We found that the noise level of the Erlenbach systems with accelerometers (GP-Acc) and with microphones (JPM) were probably affected by a new set-up of the electronic measuring and recording system in June 2016 (Fig. 5, Fig. S4). Both systems showed a higher noise level and a larger variability after the installation of the new measuring and recording

equipment. Interestingly, the noise level of the MPA system (that was put in operation in June 2016 at the Erlenbach) was





similar to that of the GP-Acc system before June 2016. These observations indicate that care must be taken when using sensors that require a constant power supply to be in recording mode, as opposed to a geophone sensor which is a passive system and where a current is induced directly by a particle impact.

We have also noted that the accelerometer sensors used in the MPA system are sensitive to changing temperature. This is visible from daily and roughly parallel fluctuations of both the temperature and noise level at the Avançon de Nant site (Fig. S6b, c) and from similar daily fluctuations of the noise level at the Erlenbach site (Fig. S6a). Furthermore, the signal of IEPE accelerometer sensors is known to be sensitive to shock impacts (Levinzon, 2015). This is illustrated in Fig. S7, that shows a temporary saturation of the signal lasting about 2 seconds, for the case of the GP-Acc system at the Erlenbach site. As a precaution, when analyzing the raw signal of the GP-Acc and MPA systems, we pre-processed it with a high-pass filter with

a threshold of 50 Hz, which partially removes the saturation effect.

A different kind of saturation may occur, if the transport rates are very high or if the signal packet after a single particle impact is of rather long duration which leads to an overlap between the two individual packets. For the JPM, the typical duration of one packet (also called "pulse" in publications by Japanese authors) is of the order of 50 to 100 ms (Mizuyama et al., 2010; Koshiba and Sumi, 2018; Choi et al., 2020). For the SPG system in contrast, the typical duration of one packet is

about 2 to 20 ms. At the Erlenbach site, the relative time occupied by packets amounted to a few percent of the total recording time during bedload sampling, even at the highest bedload transport rates $q_b$ (~1–3 kg m$^{-1}$ s$^{-1}$) (Wyss et al., 2016a). Saturation due to frequent particle impacts may be expected for the SPG only for $q_b$ clearly larger than 10 kg m$^{-1}$ s$^{-1}$. However, saturation due to frequent particle impacts may be expected for the JPM already at $q_b$ values larger than about 0.1 kg m$^{-1}$ s$^{-1}$ (Mizuyama et al., 2010).

For the MPA measurements at the Obernach flume site, we designed an algorithm for automatic detection of saturation events, which is based on the number of times the amplitude crosses the 0-line. We found that signal saturation events on MPAs were caused by particles with b-axis diameter larger than ca. 70 mm (Fig. S8a). However, the number of experiments with signal saturation is small compared to the non-saturating ones. This lets us conclude that signal saturation is not a severe constraint and occurs only for maximal amplitudes are anyway close to the boundary of the output range (Fig. S8).

This was confirmed with drop experiments using quartz spheres falling on the MPA at the Avançon. Only close to the upper limit of the accelerometer output range (500 g), signal-saturating impacts could be observed there.

### 4.4 Transport intensity, lateral signal propagation, and further observations

Previous studies with the SPG system (e.g. Wyss et al., 2016a; Chen et al., 2022; Nicollier et al., 2022b) indicated that the number and size of registered impulses primarily depend on: (i) the impact location on the plate, (ii) the particle impact

velocity or the energy of the particle impact transmitted to the plate, (iii) the particle shape and the mode of transport, and (iv) the number of particle impacts for a given size. Assuming that the geophone sensor reacts symmetrically to its centre and that a bedload particle is equally likely to impact at each point over the plate, one would expect a mean number of registered impulses for a given particle, and also the other factors contributing to the variability of the signal response would



be more and more averaged out over an increasing number of transported particles. Thus, it is not surprising that the
variability of the signal response for a given bedload mass or transport rate decreases with increasing mass (Rickenmann et
al., 2014) or increasing transport intensity (Rickenmann and Fritschi, 2017), respectively. Similarly, we can expect that the
variability of the signal response between two different impact systems will be reduced for increasing transport intensity.
This is documented by our continuous recording of *IMP* values with a 1-minute time resolution for the example of the
Erlenbach site, for which we compared GP-Acc and SPG measurements (Fig. S9) as well as measurements of two
neighbouring MPA plates (Fig. S10) for different transport intensities. The latter example (Fig. S10) also documents a
temporal shift in the relation between *IMP* (MPA04) and *IMP* (MPA03), which is also visible in the change of the mean
signal response (i.e. the $k_b$ value) over time for the calibration measurements shown in Fig. 5. Similarly, higher transport
intensities result in a stronger correlation of *IMP* values per minute, as is illustrated for the Erlenbach site when comparing
*IMP* (MPA01–04) with *IMP* (SPG07–08) (Fig. S11), *IMP* (JPM) with *IMP* (MPA01–04), and *IMP* (JPM) and *IMP* (SPG07–
08) (Fig. S12).

For the SPG system, we had identified an important factor contributing to the inter-site variability of the mean signal
response, which is both the lateral signal propagation across the steel frame structure and the longitudinal signal propagation
from concrete to the SPG array. Thereby, the magnitude of the signal propagation increases with particle size or impact
energy (Antoniazza et al., 2020; Nicollier et al., 2022a, 2022b). Given that the construction of the MPA system involves
relatively more elastomer material than in the case of the SPG system, we could expect that the signal propagation is of less
importance for the MPA system. To study the signal propagation for the MPA, we applied a modified analysis combining all
4 accelerometers of an array of 1 m width (such an array is present at the Erlenbach and Obernach sites, and two such arrays
were installed at the Avançon de Nant site) to obtain a 4-dimensional signal that evolves in time. More specifically, a time
window is started when the envelope of the signal of one of the sensors exceeds the threshold level, and the time window is
ended when all of the envelopes have dropped below the threshold level again. Within each time window, we then compared
the two maxima of the sensors that have the highest and the second highest amplitude to obtain an indication of the degree of
signal attenuation in lateral direction. For that, we took the ratio of the two maxima. For most time windows at the Avançon
de Nant site (as an example), the attenuation factor was around 30, and it tended to only slightly increase for larger particle
impacts (Fig. S13). Thus, the attenuation of the signal is clearly larger for the MPA than for the SPG system.
A preliminary comparison of the signal of the JPM and the SPG and of the GP-Acc and the SPG system was presented in
Rickenmann (2017), indicating that all the three systems provide a generally similar signal response in terms of *IMP* counts
and also in terms of maximum amplitudes (shown there for the GP-Acc and the SPG system only). In this study, we
quantitatively compared the (linear) calibration relations for the three systems. We found that both the GP-Acc, JPM, and
SPG show similar and high-quality calibration relationships (Fig. 5, Table 5), including the Albula site (Fig. S14). As
discussed above, the MPA system shows poorer calibration relationships than the SPG and GP-Acc systems (Fig. 6, Table
5). The poorer performance of the MPA may be partly due to the variability of the noise level, reflected by comparing the
ratio of minute values of JPM/SPG and MPA/SPG over time for the Erlenbach site (Fig. S15), keeping in mind that the SPG





system is a temporally stable and well performing reference system. A second reason for the poorer performance of the MPA is a larger effect of changing flow velocity on the amplitude signal response for the MPA than for the SPG system (Fig. 11),

and a third reason is the lacking sensitivity of the MPA system to sufficient signal (amplitude) response for particle sizes larger than about 76 mm (Fig. 8, 9).

For all measuring systems, we generally observed a dependency of the relative accuracy of the impulse-bedload relation on the sampled bedload mass (Fig. 6, 11). This has already been shown previously for the earlier version of the SPG system using a piezoelectric sensor instead of a geophone (Rickenmann and McArdell, 2007, 2008), and for the SPG system for

which the scatter of the data defining the calibration relation was found to decrease with increasing sampled bedload mass (Rickenmann et al., 2012, 2014; Nicollier et al., 2022b) or with increasing bedload transport rate (Rickenmann and Fritschi, 2017, Nicollier et al., 2021).

## 4.5 Preliminary experimental findings with a new prototype measuring device

As a result of the experience with several surrogate bedload monitoring systems, we have recently developed a new

prototype measuring device, the so-called Square Pipe System (SPS). This device is more compact, and cheaper in fabrication and installation costs than the SPG but is expected have a similar performance as the SPG. The prototype system is a square steel pipe, with a cross-section of 0.1 m × 0.1 m, a steel thickness of 0.008 m and a length of 1 m across the stream. It is equipped with two geophones sensors, 0.25 m and 0.75 m from one end, an accelerometer in the middle position, and a microphone attached to the closing lid at one end of the pipe.

Here we report about a preliminary analysis using a total of 287 sorted grain size experiments conducted in Obernach in summer 2021. Based on these observations, we made two types of analyses to make a direct comparison of the SPS and the SPG systems. First, we created 100 synthetic mixtures, for which we randomly selected between 2 and 20 out of the 287 sorted grain size experiments, and combined them together. For the SPS system, we only used the *IMP* values determined from the two geophone sensors in this analysis. We then determined calibration relations between *IMP* and transported

bedload mass $M$, to represent a similar range as for the field sites. Interestingly, the performance of the calibration relations is very similar for the SPG and SPS systems (Fig. 13). Second, we prepared plots showing the $k_{bj}$ values as a function of grain size $D$; $k_{bj}$ is defined similarly to equation (1), but separately for each grain size class j. From these graphs it is obvious that the two systems produce generally a similar signal response, but that the SPS is somewhat more sensible to the impact of smaller grains than the SPG (Fig. 14), which may be partly due to the somewhat more rigid structure of the SPS system.

Note that the longitudinal length (in flow direction) and the thickness of the SPS structure are significantly smaller compared to the SPG structure, resulting in a difference in the structural dynamic response under bedload particle impact.

Including also an accelerometer and a microphone sensor, apart from the two geophones, enhances the potential of the new prototype measuring device SPS to possibly better detect smaller particle sizes than the SPG system. Based on the sorted grain size experiments conducted in Obernach in summer 2021, and using the signal from all the four sensors, a machine

learning algorithm was applied to the data of the SPS system, to examine the ability to predict grain size from the recorded


Earth **Surface**
**Dynamics**
Discussions

signal. The same approach was also applied to the measuring systems SPG and MPA. A simple and a complex feature set were created from the raw signal, and out of 9 evaluated machine learning model types, CatBoost models in combination with the complex feature set performed best and achieved $R^2$-scores above 0.8 for all measuring systems (Saritas et al., 2022). This confirms the potential of the SPS that it could be used in the future to determine fractional bedload transport

rates, as it has been already demonstrated in more detail for the SPG system (Nicollier et al., 2022b).

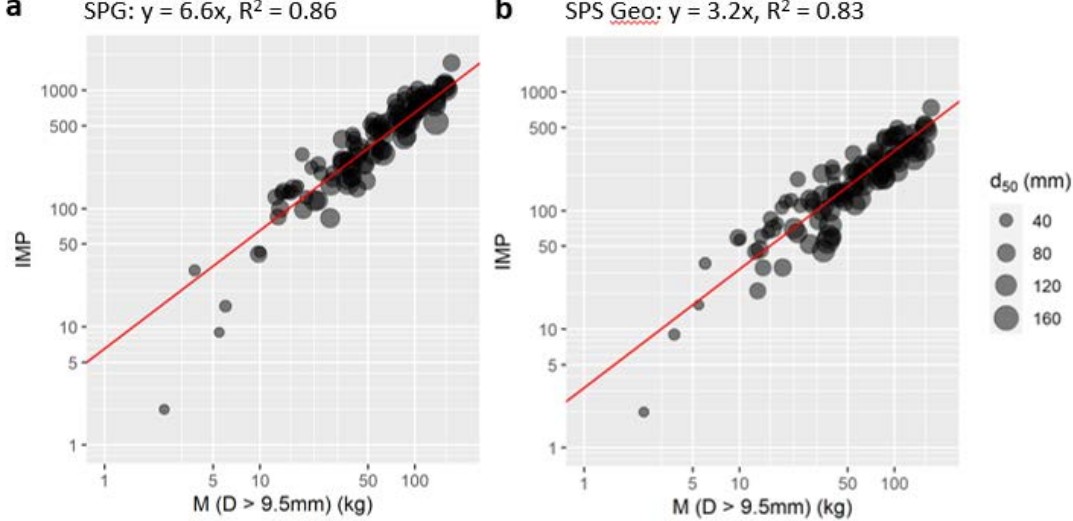

**Figure 13: Synthetic mixtures compiled from a varying number of randomly selected sorted grain size experiments in Obernach. Impulses and mass of bedload are the respective sums of the sorted grain size experiments that were selected for each synthetic mixture. (a) SPG system, (b) SPS system, geophone impulses measured with the 2 sensors.**

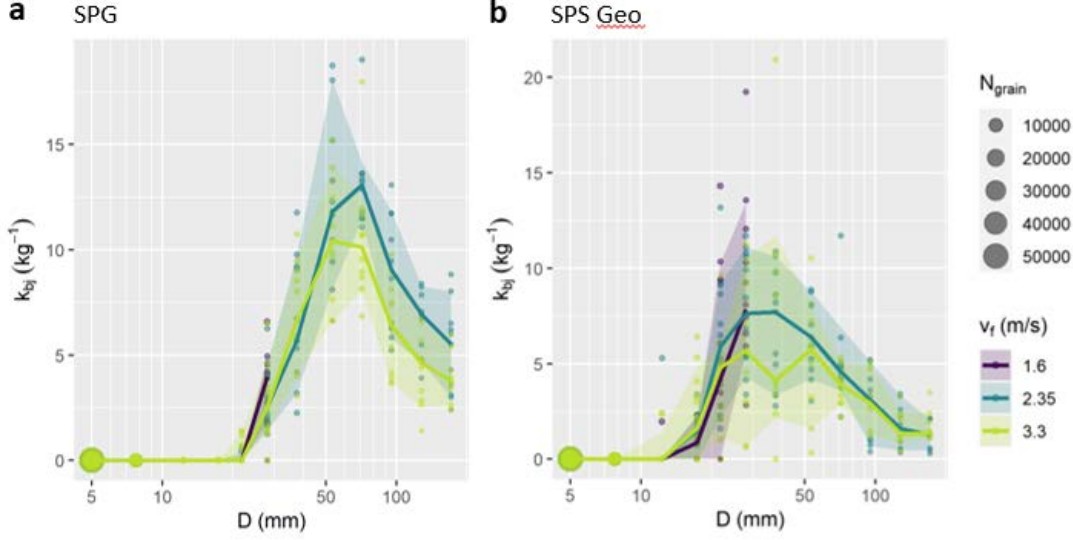

**Figure 14: Impulses per mass of bedload ($k_{bj}$) for each grain size class j, as a function of grain size $D$. Only**





**experiments from Obernach in summer 2021 for single grain size classes (sorted experiments) were used. $v_f$ is the water velocity 10 cm above the impact plates. (a) SPG system, (b) SPS system, geophone impulses measured with the 2 sensors. For the impulse counts, a threshold value $A_{min} = 0.1$ V was used for both systems.**

## 5 Conclusions

Based on calibration measurements with direct bedload samples from three field sites and an outdoor flume facility with
controlled sediment feed, we compared the performance of the SPG system with the three alternative surrogate measuring systems GP-Acc, JPM, and MPA. Our measurements at the field sites indicated that geophone sensors tend to produce a reliable signal over extended periods of time, whereas the accelerometer sensors seem to be more prone to signal instabilities. At the Albula field site, some unstable accelerometer sensors were hardly usable at all. This is presumably due to the requirement of a stable power supply, which can be difficult to maintain in field installations. If there are doubts as to
whether a reliable and high-quality installation, operation and maintenance are possible at a certain site, the use of accelerometer sensors such as the ones used in this study is to be examined with caution.

The approach of using impulse counts to approximate total bedload transport was found to work well for the SPG system. The same was true for the GP-Acc and the JPM for periods with a stable background signal, e.g. for the Erlenbach site for the period before June 2016. Applying impulse counts to approximate total bedload transport with the MPA system led to a
less accurate results. Comparing the signal response for different grain size classes, we found that the impulse count of the MPA is more sensitive to smaller grains (ca. 20–50 mm), while the SPG is more sensitive to larger grains (ca. 50–100 mm). Since the bedload samples used in this study tended to be dominated by the latter fraction, this could at least partially explain the better relationship between impulse and bedload mass for the SPG. It is reasonable to assume that the impulse–bedload relation of the MPA would be more accurate when applied to bedload material with a finer grain size distribution.
The construction of the MPA system with a relatively large volumetric proportion of elastomer material resulted in considerable dampening of the signal following the impact of larger grains. In addition, we found that the signal response of the MPA system was more variable for different flow velocities, particularly regarding the maximum amplitude and impulse counts, than that of the SPG system. As an outlook, the new and relatively cheap SPS prototype system is a more rigid structure than the MPA; the SPS is equipped with geophones, an accelerometer and a microphone, with the idea of
combining advantages of different systems. Preliminary observations at the Obernach flume site suggest that it can produce relatively stable impulse–bedload relations, and that it has a good potential for particle size identification.



**Data availability**

The dataset presented in this paper is available online on the EnviDat repository (upon final publication):
https://www.envidat.ch/#/metadata/sediment-transport-observations-in-swiss-mountain-streams.

**Author contribution**

DR, LA, TN, GA, CW, and AB conceived and planned the field measurements, DR and TN conceived and planned the flume experiments. SB and BF designed and built the data recording systems. LA, TN, GA, NS, ZC, and CW performed the measurements at the field sites and the flume. LA performed the majority of the data analysis, and wrote a first draft of the
manuscript. DR prepared the following versions, with contributions from all co-authors to the final version.

**Competing interests**

The authors declare that they have no conflict of interest.

**Acknowledgements**

We acknowledge the support of many colleagues at WSL and UNIL who helped setting up and running the bedload transport
measurements at the Erlenbach, the Albula, and the Avançon de Nant. We also appreciate the support of Arnd Hartlieb and his colleagues at the Obernach flume site of TU Munich, who helped performing the outdoor flume experiments. The study was supported by the Swiss Federal Office for the Environment (contract no. 00.5027.PZ/5563FD038), the Swiss Federal Research Institute WSL, the University auf Lausanne UNIL, and the Swiss National Science Foundation through grants 200021_137681 and 200021L_172606 to DR.

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
