# Peer review of "Comparison of calibration characteristics of different acoustic impact systems for measuring bedload transport in mountain streams"

_Earth Surface Dynamics, 2022_

## Author Comment (AC1)

**Response to the comments made by Anonymous Referee #1**

Dear Referee #1,

We thank you for your positive assessment of the manuscript. We appreciate your valuable comments that helped to improve the manuscript. We agree with most of your suggestions, and have made the modifications accordingly. Below, the reviewer comments are reported in italics, and our responses in normal font (blue color). We also appreciate your comment "I encourage the authors to continue their work trying to find cheaper and more portable ways to measure bedload".

Specific Comments

**Comment 1:** *It is unclear to me why the authors mention and include a Power Fit in Section 2 and Table 5 but then do not mention it in any of the discussions about their results.*

Response: There are two reasons why we included the power-law relations in Table 5. (i) As the correlation coefficients R are mostly quite similar for the linear and the power law-relations, indicating that a linear calibration relation provides a good description of the measurements, if the power-law exponent $\beta$ is close to one and R is fairly large (as is true for many cases in Table 5). (ii) The inclusion of the power law-relations allows a more detailed comparison with earlier studies, where also power-law calibration relations were included, based either on impulse counts or on other summary values (e.g. Rickenmann et al. 2014, Habersack et al. 2017). We inserted a related comment in the revised version of the manuscript (section 3.2) to make these points more visible.

**Comment 2:** *Lines 284-287 talk about a variability in the kb value for the SPG and MPA systems for a certain range of masses. It is unclear what this variability is: whether it is the range of kb values for each of the Obernach experiments, or whether it is the range of kb values when considering only certain size classes, or whether it is something else entirely.*

Response: In the text of lines 284-287 of the original submission, we specifically refer to Figure 8 and the observations at the Obernach experiments, thus the variability refers to the range of "individual" $k_{bi}$ values, where $k_{bi}$ is defined for each individual experiment for a given size class. This larger variability of the "individual" $k_{bi}$ values is also observed at field sites and for the mixture experiments at Obernach (Figs. 6, 11). The variability of the $k_{bi}$ values is reflected by the position of each data point relative to a linear mean trend line in a Figure of IMP values vs. bedload mass M. We explain this issue in a clearer way in the revised version of the manuscript.

Technical Comments

*Line 40 – remove the word "this"*

Response:      corrected

*Line 73 – change "to detect" to "the detection of" or similar*

Response:      corrected

*Line 74 – remove the word "indicate"*

Response:      corrected

*Line 148 – change "the full raw signal is" to "the full raw signal was"*

Response:      corrected

*Figure 6 – The legend entries are in boxes of different sizes. This can lead the reader to think that these sizes are significant. In addition, the light green colored equation is very difficult to read*

Response:      Figure 6 has been revised considering the comments made by the reviewer.

*Lines 268-269: I cannot make sense of this statement. Suggest re-wording for clarity.*

Response:      We agree that the second part of the sentence was unclear. This part was deleted, and the first part was modified for better clarity (L292-294 in the "tracked-changes" version of revised manuscript).

*Line 271 – remove the words "and", "if" so the it reads "For all four sites (for the Erlenbach…"*

Response:      corrected

*Line 282 – begins a long and complicated sentence. I suggest breaking it into two sentences "… by larger particle impacts. This is evidenced…."*

Response:      corrected

*Line 296 – remove the word "relatively"*

Response:      corrected

**Line 317** – *remove one of the "that"s*

Response:      corrected

**Line 346** – *remove the comma*

Response:      corrected

**Line 352** – *remove the word "clearly"*

Response:      corrected

**Line 353** – *remove the word "already"*

Response:      corrected

**Line 359** – *needs to be reworded. For example "… occurs only for maximum amplitudes that are close …."*

Response:      corrected

**Line 361** – *needs to be reworded. For example "… range (500 g) were signal-saturating impacts observed."*

Response:      corrected

**Line 379** – *The last "and" should be "with" to match the rest of the sentence*

Response:      corrected

**Line 383** – *remove the word "Thereby,"*

Response:      corrected

**Line 394** – *remove the word "clearly"*

Response:      corrected

**Line 428** – *remove the word "generally" and change the word "sensible" to "sensitive"*

Response:      corrected

***Line 434*** *– remove the word "the" between "all" and "four"*

Response:     corrected

***Line 435*** *– remove the phrase "to the data of the SPS system"*

Response:     corrected

***Line 448*** *– remove the phrase "at all"*

Response:     corrected

***Line 454*** *– remove the word "a" after "MPA system led to"*

Response:     corrected

***Line 457*** *– remove the phrase "at least"*

Response:     corrected

***Line 465*** *– I suggest ending the sentence instead of using a semi-colon*

Response:     corrected

**Further changes**

We have also made some further minor changes to the original manuscript. These mainly concern typos, updating references of some recently published papers, and general rewording of terms or sentences. All changes can be found in the "tracked-changes" version of the manuscript.

---

## Author Comment (AC2)

**Response to the comments made by Anonymous Referee #1**

Dear Referee #2,

We thank you for your positive assessment of the manuscript. We appreciate your valuable comments that helped to improve the manuscript. We agree with most of your suggestions, and have made the modifications accordingly. Below, the comments are reported in italics, and our responses in normal font (blue color).

General Comments

***Comment 1:*** *I am unsure what the purpose of the IQA measurements are – the authors should elaborate more on how this was used in the analyses.*

Response: The definition of the IQA values is given in L168-170 (first version of manuscript), and in L182-183 it is explained how an average noise level (per minute) is calculated from the IQA values. These average noise levels are shown in Figure 4. The caption of Figure 4 was slightly modified to make this point clearer.

***Comment 2:*** *In the methods section, the authors describe fitting both linear and power relations to the data, but only show results from the linear models subsequently. Is this because the power relations were not as strong? In this case, I would consider removing from the article or mentioning their inferior performance somewhere in the results.*

Response: There are two reasons why we included the power-law relations in Table 5. (i) As the correlation coefficients R are mostly quite similar for the linear and the power law-relations, indicating that a linear calibration relation provides a good description of the measurements, if the power-law exponent β is close to one and R is fairly large (as is true for many cases in Table 5). (ii) The inclusion of the power law-relations allows a more detailed comparison with earlier studies, where also power-law calibration relations were included, based either on impulse counts or on other summary values (e.g. Rickenmann et al. 2014, Habersack et al. 2017). We inserted a related comment in the revised version of the manuscript (section 3.2) to make these points more visible.

***Comment 3:*** *Section 4.5 and the results of the SPS system seem to come out of the blue and are a bit awkward in my opinion to be presented here. I wonder if the authors consider placing this as a main objective, or at minimum introducing this system earlier.*

Response: We added a schematic sketch of the SPS in Figure 2, and we introduced this system now in section 2.1. However, we do not see the SPS as being a main objective of the study since we obtained only some preliminary calibration measurements so far.

Specific Comments

***Line 40*** *– Remove the "this"*

Response:      corrected

***Line 74*** *– Missing parentheses bracket here.*

Response:      corrected

*In **Figure 1** it appears that the acoustic measuring systems are installed within a concrete liner – could the authors elaborate more on their installation? Is this the nature of the stream in this location, or is the concrete slab placed directly within the streambed?*

Response:      At the location of the measuring devices, the stream reach is not natural. When the sediment retention basin was built, an artificial approach flow channel was constructed upstream of this basin. We added more information on the artificial approach flow channel in the caption of Figure 1.

***Line 128-129:*** *How are the authors sure that the metal frames are robust against these forces? Are the previous studies or measurements confirming this?*

Response:      Our main experience is with the SPG construction, which has never been damaged at any of the more than 20 sites where this device was installed. The dimensions of the MPA steel box were roughly scaled from the SPG structure. So far, no damage occurred to the MPA structures. We are aware of damage to the JPM pipe at some locations (likely due to the impact of large boulders), which has a wall thickness of 3 mm only.

***Line 149-151:*** *Can the authors elaborate more on how they chose the recording frequency for each system?*

Response:      We include now more information on this issue in section 2.2. The new text reads: "During a calibration measurement, i.e. the time period of direct bedload sampling, the full raw signal was recorded for each measuring system (Table 2). The geophone sensor we used was designed for seismic applications, and may not yield reliable measurements for frequencies larger than a few kHz. Therefore, we decided to use a measuring frequency of 10 kHz for the SPG system at all sites. Microphones and accelerometers are able to pick up higher frequencies, and therefore we decided to use a measuring frequency of initially 50 kHz for the JPM and 20 kHz for the MPA at the Erlenbach (Table 2). Due to limitations of the data acquisition systems at other sites, we used a measuring frequency of 10 kHz for the GP-Acc at the Albula and for the MPA at the Avançon stream. For the JPM and the MPA at the Erlenbach, the raw signal was down-sampled to 10 kHz before further processing for the calibration analysis, to avoid any possible bias due to differences in sampling frequency when comparing the MPA measurements from the Erlenbach and the other sites."

***Table 2:*** *For units, is V = volts and g = ?*

Response:    Yes, this was added in the caption of Table 2.

*Line 166: Check that A min is showing up correctly as a subscript.*

Response:    corrected

*Line 189: Can the authors explain what the centroid frequency is?*

Response:    We added a definition in the second to last paragraph of section 2.2 ("calculated as the weighted mean of the frequencies present in the signal, determined using a Fourier transform").

*Line 246-249: Is this change in calibration relation also attributed to the change in the cable?*

Response:    The cable from the sensor to the measuring system was not changed, but the data acquisition system was changed.

*Figure 5b: Is M shown here independently measured bedload mass?*

Response:    Yes. (The methods for bedload sampling are summarized in Table 4.)

*In Figure 5b, it is difficult to distinguish the 'before June 2016' and 'after June 2016' points, particularly for the JPM system, for which it is important. I would consider using a different colour to make it more visible.*

Response:    The readability of the figure was improved.

*Line 259: This is the first mention of this fixed rough bed upstream – I would like to see more elaboration on this set up.*

Response:    We added a second paragraph in section 2.3 that reads: "At the outdoor flume facility in Obernach, the bed slope and bed roughness of the Albula and Avançon de Nant field sites were reconstructed in a 24 m long and 1 m wide test reach (Nicollier et al., 2021). The part of the reach upstream of the surrogate measuring devices consisted of a paved section, where pebbles with a characteristic size of $D_{67}$ and $D_{84}$ of the surface bed material were embedded in concrete ($D_{xx}$ refers to the grain size for which xx percent of the particles are finer), to provide a similar roughness as at the field sites. In each experiment, sediment particles of known sizes were fed into the flume sufficiently far upstream of the measuring devices so that they were transported along the bed."

*Line 281: I would choose either "generally" or "clearly" to simplify the sentence.*

Response:     corrected

***Figure 12:*** *The authors should check to make sure the figure titles and axes labels show up without the spelling error red lines in this figure and in other figures (I believe the issue is in Figures 13 and 14 as well)*

Response:     corrected

***Line 346:*** *No need for a comma after "occur".*

Response:     corrected

**Further changes**

We have also made some further minor changes to the original manuscript. These mainly concern typos, updating references of some recently published papers, and general rewording of terms or sentences. All changes can be found in the "tracked-changes" version of the manuscript.